# MULTIPLE SEQUENCE ALIGNMENT AS A SEQUENCE-TO-SEQUENCE LEARNING PROBLEM

**Edo Dotan**[1‡]**, Yonatan Belinkov**[2†*]**, Oren Avram**[3]**, Elya Wygoda**[1]**, Noa Ecker**[1]**, Michael Alburquerque**[1]**, Omri Keren**[1]**, Gil Loewenthal**[1]**, and Tal Pupko**[1†]

[1] Tel Aviv University
[2] Technion – Israel Institute of Technology
[3] University of California Los Angeles
‡ `edodotan@mail.tau.ac.il`
† Corresponding authors: `belinkov@technion.ac.il, talp@tauex.tau.ac.il`

## ABSTRACT

The sequence alignment problem is one of the most fundamental problems in bioinformatics and a plethora of methods were devised to tackle it. Here we introduce BetaAlign, a methodology for aligning sequences using an NLP approach. BetaAlign accounts for the possible variability of the evolutionary process among different datasets by using an ensemble of transformers, each trained on millions of samples generated from a different evolutionary model. Our approach leads to alignment accuracy that is similar and often better than commonly used methods, such as MAFFT, DIALIGN, ClustalW, T-Coffee, PRANK, and MUSCLE.

## 1 INTRODUCTION

A multiple sequence alignment (MSA) provides a record of homology at the single position resolution within a set of homologous sequences. In order to infer the MSA from input sequences, one has to consider different evolutionary events, such as substitutions and indels (i.e., insertions and deletions). MSAs can be computed for DNA, RNA, or amino acid sequences. MSA inference is considered one of the most common problems in biology (Van Noorden et al., 2014). Moreover, MSAs are required input for various widely-used bioinformatics methods such as domain analysis, phylogenetic reconstruction, motif finding, and ancestral sequence reconstruction (Kemena & Notredame, 2009; Avram et al., 2019). These methods assume the correctness of the MSA, and their performance might degrade when inaccurate MSAs are used as input (Ogden & Rosenberg, 2006; Privman et al., 2012).

MSA algorithms typically assume a fixed cost matrix as input, i.e., the penalty for aligning non-identical characters and the reward for aligning identical characters. They also assume a penalty for the introduction of gaps. These costs dictate the score of each possible alignment, and the algorithm aims to output the alignment with the highest score. Previous studies demonstrated that fitting the cost matrix configuration to the data increases the MSA inference accuracy (Rubio-Largo et al., 2018; Llinares-López et al., 2021). Thus, MSA algorithms often allow users to tune parameters that control the MSA computation. However, in practice, these parameters are seldom altered, and only a few default configurations are used.

Alignment algorithms are often benchmarked against empirical alignment regions, which are thought to be reliable. However, such regions within alignments do not reflect the universe of alignment problems (Thompson et al., 1994). Of note, these regions were often manually computed, so there is no guarantee that they represent a reliable "gold standard" (Morrison, 2009). When testing alignment programs with simulated complex alignments, the results differ from the benchmarks results (Chang et al., 2014).

---

*Supported by the Viterbi Fellowship in the Center for Computer Engineering at the Technion.

In the last decade, deep-learning algorithms have revolutionized various fields (LeCun et al., 2015), including computer vision (Voulodimos et al., 2018), natural language processing (NLP) (Young et al., 2018), sequence correction (Baid et al., 2021), and medical diagnosis (Rakocz et al., 2021; Hill et al., 2021). Neural-network solutions often resulted in a substantial increase in prediction accuracy compared to traditional algorithms. Deep learning has also changed molecular biology and evolutionary research, e.g., by allowing accurate predictions of three-dimensional protein structures using AlphaFold (Jumper et al., 2021).

We propose BetaAlign, a deep-learning method that is trained on known alignments, and is able to accurately align novel sets of sequences. The method is based on the "transformer" architecture (Vaswani et al., 2017), a recent deep-learning architecture designed for sequence-to-sequence tasks, which was originally developed for machine translation. BetaAlign was trained on millions of alignments drawn from different evolutionary models. Our analyses demonstrate that BetaAlign has comparable, and in some cases superior, accuracy compared to the most popular MSA algorithms: T-Coffee (Notredame et al., 2000), ClustalW (Larkin et al., 2007), DIALIGN (Morgenstern, 2004), MUSCLE (Edgar, 2004), MAFFT (Katoh & Standley, 2013) and PRANK (Löytynoja & Goldman, 2008).

BetaAlign converts the alignment problem into a sequence-to-sequence learning problem, a well-studied problem within the NLP field (Hirschberg & Manning, 2015). We first present the NLP-based approach for pairwise alignment. We represent both the unaligned sequences and the resulting alignment as sentences in two different "languages". The language of the input sequences (the source language) is termed Concat, because, in this representation, each pair of input sequences is concatenated with the pipe character ("|") representing the border between the sequences. In this language, each character in the resulting string is considered a different token. For example, when we aligned the sequence "AAG" against the sequence "ACGG", in the Concat language, this will be represented by the sentence "A A G | A C G G" (Fig. 1). In this sentence, there are eight tokens. Of note, in the Concat language there are only five possible unique tokens. A dictionary of a language is the entire set of tokens used in that language, and thus in the Concat language, the dictionary is the set {"A", "C", "G", "T", "|"}.

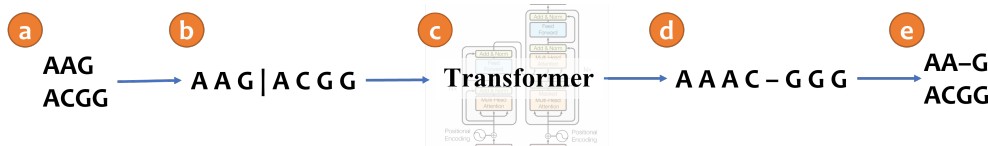

Figure 1: An illustration of aligning sequences with sequence-to-sequence learning. (a) Consider two input sequences "AAG" and "ACGG". (b) The result of encoding the unaligned sequences into the source language (Concat representation). (c) The sentence from the source language is translated to the target language via a transformer model. (d) The translated sentence in the target language (Spaces representation). (e) The resulting alignment, decoded from the translated sentence, in which "AA-G" is aligned to "ACGG". The transformer architecture illustration is adapted from Vaswani et al. (2017).

The target language, i.e., the language of the output alignment, is termed here Spaces. In this language, the dictionary is the set {"A", "C", "G", "T", "–"}. In this representation, the tokens of the two different aligned sequences are interleaved. Thus, the first two tokens in the Spaces output sentence are the first character of the first sequence and the first character of the second sequence, respectively. The third and fourth tokens correspond to the second column of the pairwise alignment, etc. In the above example, assume that in the output alignment "AA-G" is aligned to "ACGG", then, in the Spaces language, this will be represented as the sentence "A A A C – G G G" (Fig. 1). We have compared the performance of different source and target representations (see Section A.2.2).

With these representations, the input sequences are a sentence in one language, the output alignment is a sentence in another language, and a perfect alignment algorithm is requested to accurately transduce one language to the other. The details regarding the training and execution of the NLP transformers are provided in Section 2, and Section A. Of note, the occasional generation of invalid alignments was encountered while developing BetaAlign. This challenge was addressed by introducing an ensemble of transformers (see Section A.2.3).

## 1.1 RELATED WORK

The Needleman & Wunsch (1970) algorithm to infer pairwise alignment assumes an affine gap penalty to indels, which is equivalent to assuming that the indel length distribution is geometric. The cost of substitutions and the reward for exact matches are equivalent to assuming specific probabilities for character substitutions. Assuming that this model captures the evolutionary dynamics of all datasets is unrealistic because: (1) These simple assumptions are known to be violated in empirical datasets, e.g., the probabilities of proteins substitutions depend on the context. For example, whether the residue is exposed to the solvent or not; (2) It was previously shown that indel lengths are not geometrically distributed in protein datasets (Benner et al., 1993; Chang & Benner, 2004; Zhang et al., 2010). Different datasets differ in their evolutionary dynamics, and thus the same model should not be applied to all datasets. Indeed, it was shown that fitting the scoring schemes to the specific dataset can improve the performance of the dynamic programming algorithms for alignment inference, suggesting that variability among datasets is common (Llinares-López et al., 2021). These observations motivate the development of aligners that are able to learn data-specific attributes.

One could generalize this dynamic programming to multiple sequences with the time complexity of $O(L^N)$ (where $L$ is considered the sequence length and $N$ is the number of sequences), and thus, MSA computation becomes quickly impractical. Today multiple-sequence aligners are mostly based on the progressive method in which pairwise alignments are iteratively combined to infer the multiple sequence alignment (e.g. Löytynoja, 2014; Katoh & Standley, 2013; Edgar, 2004).

## 2 MATERIALS AND METHODS

### 2.1 AN NLP-BASED APPROACH

Our hypothesis is that cutting-edge NLP tools can learn from known examples, and a trained model can infer accurate alignments for unseen data. A plethora of neural network architectures is available in the NLP domain for transduction, including recurrent neural networks (Sutskever et al., 2014) and transformers (Vaswani et al., 2017). As these architectures are sequence-to-sequence, the input to these models is a sentence, and the output is a predicted transduction. Our approach relies on optimizing the model based on training data, i.e., the learning phase of the algorithm. In our case, a massive amount of training data, i.e., true alignments, is obtained using simulations. We simulate the training alignments using SpartaABC (Loewenthal et al., 2021). SpartaABC allows simulating datasets along a phylogenetic tree, with various indel-length distributions, and with different indel-to-substitution rate ratios. It allows finding indel model parameters that best describe a specific empirical dataset. It also assumes that the evolutionary dynamics of insertions and deletions are characterized by a different set of parameters, i.e., a rich-indel model (Loewenthal et al., 2021).

Formally, let $x$ denote the input sequences and $y$ denote the correct MSA. Each training data point is a pair $(x, y)$. During the training phase, the NLP algorithm learns a function $f_\theta$ that maps $x$ to $y$, i.e., a transformer. This function should minimize the discrepancy between $y$ and $f_\theta(x)$. The function $f_\theta$ depends on a set of tunable parameters, $\theta$, which defines the architecture and the internal representation of the transformer. Given an unseen test sequence, which was not used for training, the predicted alignment is obtained by applying $f_\theta$ to this input.

Note that the values of the pairs $(x, y)$ depend on an evolutionary model $\psi$ that includes the topology and branch lengths of the phylogenetic tree, as well as the substitution and indel model parameters. Changing $\psi$ would alter the training data distribution, which in turn, should lead to a different $f_\theta$. It is possible to train the model on data from a specific $\psi$, or train a model in which $\psi$ is sampled from a prior distribution over its model parameters.

If we apply the `Concat` encoding with the `Pairs` decoding, the source dictionary size is the set of five characters and the size of the target dictionary is the set of 24 characters (see Section A.1.8). In NLP transduction, each token in each dictionary has to be embedded in a higher dimensional space. Thus, two different embeddings are needed (for the source and target dictionaries). In contrast, we can consider a joint dictionary for applying the `Concat` or `Crisscross` (see Section A.1.8) encodings with the `Spaces` decoding. The joint dictionary of `Concat` and `Spaces` has six tokens: "A", "C", "G", "T", "–", "|". We note that the token "–" is never found in the input, while the token

"|" is never found in the output. Nevertheless, we assume a joint dictionary for both the input and the output. The joint dictionary of `Crisscross` and `Spaces` has five tokens: "A", "C", "G", "T", "–". Thus, the same embedding function is used for both the source language, i.e., the unaligned sequences, and the target language, i.e., the aligned sequences. One of the advantages of the joint dictionaries over the separate dictionaries of `Concat` with `Pairs`, is that the dictionaries have a fixed size regardless of how many sequences are included in the alignment (in the `Pairs` representation, we should use triplets for alignments with three sequences, quadruplets for alignments with four sequences, etc.) Thus, we only use the `Spaces` language when we generalize from pairwise to multiple alignment.

## 2.2 SPLITTING THE UNALIGNED SEQUENCES TO SEGMENTS

The transformers we applied for aligning sequences are limited to sentences of up to 1,024 tokens, which usually suffices for natural languages. When aligning biological sequences, the sentence that represents the unaligned sequences is often much longer than this threshold, and the transformers fail to transduce it due to memory and run-time limitations (both these factors are proportional to the square of the sequence length). We exemplify our approach for aligning long sequences on an example comprising five sequences, each of which of length of at least 2,000 characters. In our "segmentation" methodology, we first define a parameter that we call segment size, which depends on the number of sequences. In our example of five sequences, we selected a segment size of 110 characters. Thus, when we start the alignment process, we first process 554 tokens (110 characters from each sequence and four "|" characters, when using the `Concat` representation).

When aligning segments, we use dedicated transformers for this task that we call "segmented transformers". The segmented transformers share the same architecture with the standard transformers, but they are trained on different data. In addition, while standard transformers were designed to always align the entire input sentence, segmented transformers align only prefixes of the unaligned sequences. Specifically, the segmented transformers were designed so that the last column in the resulting aligned sequences does not include gaps.

To generate the training data for the segmented transformers, we simulated long alignments (see Section A.1.2). This provided us with long unaligned sequences, which we call $X$, and "true" MSA, which we call $Y$. The training data for the segmented transformer are pairs of $(x, y)$, in which $x$ represents a segment of the unaligned sequence taken from $X$, and $y$ represents a subregion of $Y$ that corresponds to $x$. Both $x$ and $y$ are short enough to allow training a transformer.

To divide $X$ and $Y$ to multiple short $(x, y)$ pairs, we repeated the following process. We first extract the first 110 characters from each of the five sequences, which will be the unaligned segment, i.e., the $x$. In the true alignment, we find the column with the highest index so that the number of non-gap characters of each sequence to the left of this column is smaller or equal to the segment size. Once this column is found, we inspect if it has any indels. If so, we inspect the previous column. We repeat this process until we find a column without indels. The obtained (first) column without indel characters marks the "true" alignment for this segment, i.e., the $y$. Hence, the source sentence is the first 110 characters of each unaligned sequence ($x$)

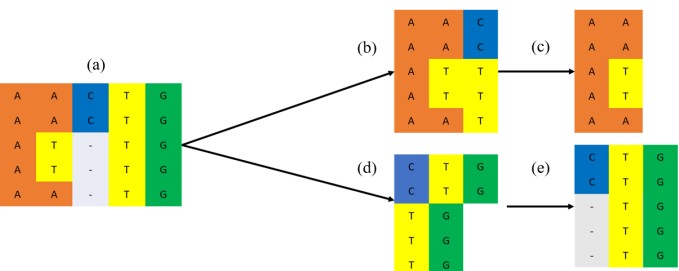

Figure 2: Creating training data for segmented transformers, explained using a segment size of three characters. (a) The true alignment; (b) The unaligned sequences of the first segment, i.e., the first $x$ as described in the Section 2.2; (c) The segment of the true MSA that corresponds to the $x$ presented in b, i.e., the first $y$. Note that the third column of the true MSA was not selected as it contains gaps, and the forth column was not selected, as it includes four characters of some of the original sequences (i.e., it is longer than the segment size of three); (d) The unaligned sequences of the second segment; (e) The corresponding segment in the true alignment.

and the target is the "true" alignment ($y$) from the first column until the indel-free column that was found. From both $X$ and $Y$, we remove all the characters that appear in $y$ (the characters that were successfully aligned). This process is repeated with the new $X$ and $Y$ until they are empty, i.e., they were divided to multiple $(x, y)$ pairs. Of note, when analyzing the last segments, it is possible that fewer than 554 tokens are translated. For explanatory purposes, we assume a segment size of three and five input sequences (Fig. 2). Simply diving $X$ and $Y$ to pairs of arbitrary length introduces various problems as illustrated in Section 2.3.

The above procedure describes the generation of training data, for which the true alignment is known. We now turn to describe how to run a trained model. Here, we first transduce the first segment. However, in order to decide which characters to remove and which to retain, we cannot rely on the identification of an indel-free column in the true alignment as we did for the training data (since we do not know the true alignment). Thus, we analyze the resulting alignment, and mark which characters of the input segments were translated from each sequence. These characters are removed, and the process is repeated. This iteration stops when all characters from the unaligned sequences were processed. Of note, different trained transformers may divide the segments differently. After aligning all of the segments, we reconstruct the resulting alignment by concatenating the resulting alignment of each segment.

## 2.3 PROBLEMS ARISE WHEN SIMPLY SPLITTING MSAS

The logic for searching for an indel-free column is illustrated in Fig. 3, i.e., an example showing the problems without indel-free last column. For explanatory purposes, we assume a segment size of three. The algorithm starts by translating the first three characters from each of the five sequences (Fig. 3b). The indel in the last column of the first segment (Fig. 3c) introduces new complexity, as this indel has a perfect match in the next segment (Fig. 3e). In our approach, this is solved by training segmented transformers. If we choose to use indels in the last column, we move the complexity of the segment approach into the concatenation of the

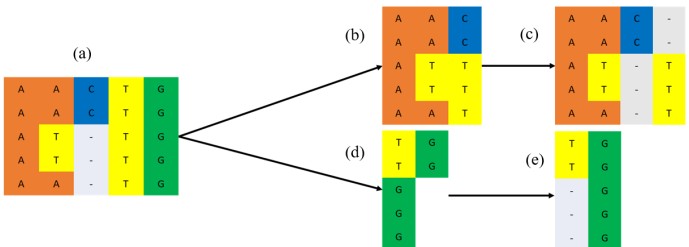

Figure 3: Importance of indel-free last column of segments. Consider (a) to be the true alignment, with a segment size of three characters. Panels (b) and (c) refer to the unaligned sequences and the aligned results, of the first segment respectively. Similarly, panels (d) and (e) refer to the unaligned sequences and the aligned results of the second segment, respectively. Note the difference between (a) and the concatenation of the results, as the fourth and fifth columns of the latter should be the fourth column in (a).

segments afterward. One will need to make sure the indels at the resulted alignment make sense. Of note, this approach makes each of the indels appear in only one segment (but a segment may harbor multiple indels).

## 3 RESULTS

### 3.1 COMPARING PERFORMANCE TO DIFFERENT ALIGNERS

We compared the performance of BetaAlign to the state-of-the-art alignment algorithms: ClustalW (Larkin et al., 2007), DIALIGN (Morgenstern, 2004), MAFFT (Katoh & Standley, 2013), T-Coffee (Notredame et al., 2000), PRANK (Löytynoja & Goldman, 2008), and MUSCLE (Edgar, 2004). For each number of sequences from two to ten, the performance was compared on a simulated test dataset comprising 3,000 nucleotide MSAs (Fig. 4a). BetaAlign had the lowest error rate for any number of aligned sequences (paired t-test; $p < 10^{-20}$). For all alignment methods, the error increases as the number of sequences increases. Notably, ClustalW and DIALIGN were much more affected by the increase in the number of input sequences, compared to other methods. Of note, this analysis was conducted by applying the `Concat` source language with the `Spaces` target language.

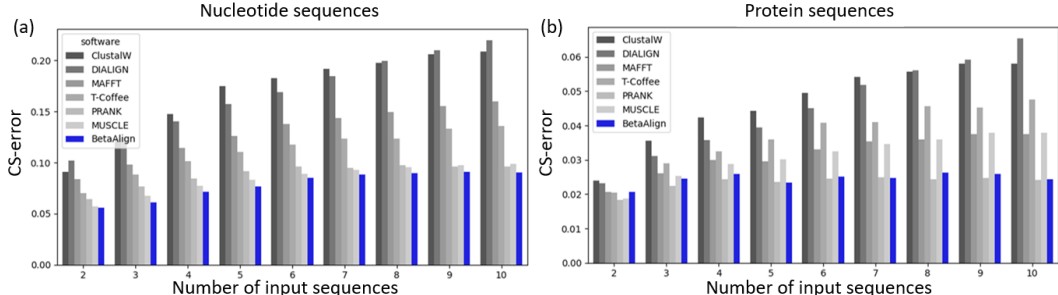

Figure 4: Comparing the accuracy of BetaAlign to other aligners on the SND1 nucleotide dataset (a) and on the SPD1LG protein dataset (b). The lower the column score error (CS-error), the more accurate is the aligner (see Section A.1.3).

We repeated the same analysis for the protein dataset. The error of all methods was lower compared to the nucleotide sequences, most likely reflecting the increase in alphabet size, which helps identify homologous regions (Fig. 4b). PRANK and BetaAlign outperformed all other aligners. As seen from Fig. 4, there is no single optimal method to align all types of sequences.

## 3.2 GENERALIZING TO LARGE MSAS

Aligning long sequences with transformers introduces new challenges. The attention map calculated inside the transformer is $O(n^2m^2)$, both in term of memory and running time, where $n$ and $m$ are the number of sequences and the sequences' length, respectively. Note, that $mn$ reflects the total tokens that need to be processed. Thus, increasing the alignment size causes memory and time problems. This is a known problem in the NLP domain, and different efficient transformers were created in order to tackle this issue (Lin et al., 2021). Those architectures cannot be directly applied to the alignment problem (see Section 3.5). Hence, we developed a new way to align longer sequences by splitting the input sequences into short segments and training the transformers to align each segment (the size of the segments varies along the run, but is approximately 100 alignment columns). We compared the performance of all aligners on MSAs of five sequences, with true alignment lengths up to 2,500 bases and 2,508 amino acids, for nucleotide and protein sequences, respectively. Clearly, the performance of BetaAlign is affected by the division of sequences into short segments, both when aligning nucleotide (Fig. 5a) and protein (Fig. 5b) sequences. Notably, even with this approximation, BetaAlign is more accurate than the widely used aligners MAFFT and ClustalW for nucleotide sequences and more than MAFFT, ClustalW, and MUSCLE for protein sequences.

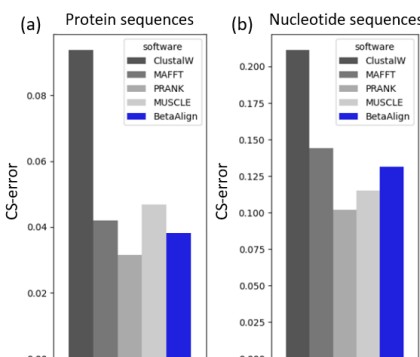

Figure 5: Performance on MSAs with five (a) protein and (b) nucleotide sequences. The average MSA length was 760 bases and 765 amino acids, for nucleotide and protein sequences, respectively. These datasets were aligned after dividing the input into short segments, thus overcoming computational memory and run time limitations introduced by large MSAs.

## 3.3 BETAALIGN CAN BE TRAINED TO CAPTURE SPECIFIC DATA ATTRIBUTES

In this section, we describe two experiments that demonstrate the importance of fitting the scoring scheme to specific datasets. Evolutionary processes generating sequence data substantially vary across the tree of life and among different regions within a genome. To this end, BetaAlign was trained on simulated data which were generated based on prior assumptions, which characterize the evolutionary dynamics of indels in Drosophila (Loewenthal et al., 2021). Fig. 6a displays the results

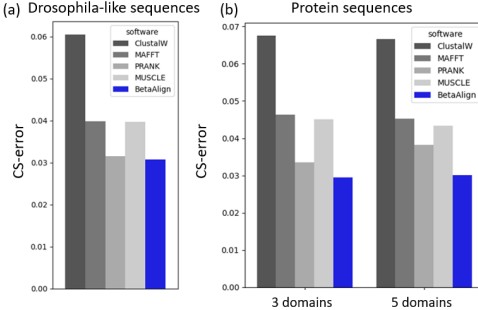

Figure 6: Evaluating the different aligners on datasets that reflect different scenarios found in biological sequences. (a) Drosophila-like datasets with high deletion rates. (b) Protein-like datasets comprised of either three or five domains, each domain with its own indel-evolutionary dynamics.

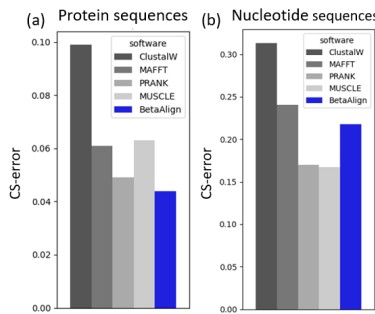

Figure 7: The effect of model misspecification on the accuracy of protein (a) and nucleotide (b) datasets. Train and test alignments had 10 sequences. Spatial variation was only introduced to the test data, i.e., different regions of the alignment were simulated with different sets of indel model parameters.

of the different aligners. BetaAlign and PRANK (Löytynoja & Goldman, 2008) outperformed all of the other aligners (paired t-test; $p < 10^{-3}$). We hypothesize that the high accuracy of PRANK stems from its ability to differentiate insertions and deletions and the fact that deletions occur at higher rates than insertions in this Drosophila-like dataset.

Next, we tested the performance of the different aligners on proteins containing different domains, e.g., with different electrochemical properties. Each of the domains has its own indel rates and distributions, reflecting the biological phenomenon of variation in the evolutionary dynamics among different regions within a single protein. We simulated proteins with either three or five domains and compared the accuracy of the various aligners (Fig. 6b). BetaAlign outperformed all other aligners, clearly demonstrating the ability of our approach to learn complicated indel patterns from training data (paired t-test; $p < 10^{-5}$). Interestingly, increasing the number of domains led to significant loss in PRANK's inference accuracy, suggesting that it is more sensitive to spatial variation in indel processes within the analyzed sequences. We conducted another analysis characterizing the errors of the different aligners (see Section A.2.1).

## 3.4 MODEL MISSPECIFICATION

We next evaluated the sensitivity of BetaAlign to model misspecifications. Specifically, we trained BetaAlign on alignments without spatial variations, but tested its performance on alignments generated with spatial variations. Test datasets were created by generating four "segments", each with a different indel dynamics (see Section A.1.11). BetaAlign was substantially more accurate on the test protein dataset compared to other aligners (Fig. 7a). On the nucleotide dataset, BetaAlign CS-error (see Section A.1.3) was lower than two aligners but higher than the other two (Fig. 7b). We note that without model misspecification, BetaAlign had the highest accuracy compared to all other aligners on nucleotide datasets and was not the best for protein datasets. Model misspecification changed this order, as BetaAlign is now the most accurate aligner for protein dataset and had intermediate accuracy on nucleotide dataset. In general, machine-learning models are sensitive to deviations between the training and testing distributions. We found it surprising as in this case, the classic aligners on the protein datasets were more sensitive than BetaAlign.

## 3.5 ATTENTION MAP COMPARISON

BetaAlign harnesses NLP-based sequence-to-sequence methods to analyze biological data. However, it is clear that biological data are substantially different from natural languages data, e.g., they have a much smaller dictionary size (six for nucleotides using the `Spaces` representation, compared to an order of a hundred thousand tokens in vocabularies of natural languages). Attention maps provide a gateway for understanding the internal properties of various transformers. Put simply, attention maps show on which parts of the sequences the transformer focuses while generating

the next token. Fig. 8a displays the averaged attention map gathered from the attention layers of the decoder (in this case, when processing an amino-acid MSA of seven sequences). Fig. 8b shows an attention map from an NLP English-to-French translation task (Belinkov & Glass, 2019; Bahdanau et al., 2016).

Note that each attention map corresponds to translation of a single sentence. In the case of Fig. 8b, the translation is of the sentence "The agreement of the European Economic Area was signed in August 1992.". The sentence translated in Fig. 8a is a concatenation of seven unaligned sequences of length approximately 40 amino-acids. Clearly, natural language sentences are much shorter, on average, than sentences processed by BetaAlign, and thus, the BetaAlign attention map is much larger (consisting of hundreds of tokens) than the natural language one (consisting of dozens of tokens). Moreover, the attention in the natural languages is focused on one diagonal while the attention in BetaAlign is focused on several diagonals, the number of which equals the number of inputs sequences. The difference stems from the fact that in the BetaAlign sequence-to-sequence task, generating the next token requires assigning more weight to information that is distributed among all sequences. Of note, this analysis was conducted on the `Concat` and `Spaces` for the source and target languages, respectively.

Fig. 8c is an enlargement of the attention map around the rightmost diagonal of Fig. 8a. The x-axis refers to the characters of the input sentence and the y-axis refers to the translated characters of the output sentence. As can be seen, the diagonal is actually a step of seven rows and one column. This is the logical explanation as each of the seven rows extracts information from this column, i.e., the characters of a column at the resulting MSA are all dependent on the same character in the unaligned sequences. The red arrows mark the corresponding output and input characters. As can be seen, the weight is higher in this area, i.e., the transformer learned the importance of the corresponding area.

Due to the dynamics of our attention maps (specifically, the focus split), we cannot easily use NLP architectures for longer sentences, which have dedicated attention maps (Lin et al., 2021). In our segmentation strategy, instead of calculating the entire attention map, we focus the attention on specific places of the diagonals. Future research will focus on improving attention calculations.

## 4 DISCUSSION

We have designed and implemented a new technology that aligns sequences with accuracy that competes with state-of-the-art methods. Our deep learning-based approach is fundamentally different from current algorithms, which rely on conventional scoring schemes. Our method allows us to personalize the alignment process for inputs that differ in their evolutionary dynamics. The increased knowledge regarding the differences in mutation processes and selection regimes among different species allows us to easily train BetaAlign on clade-specific datasets, i.e., we are capable of creating different aligners per kingdom and even per species. Similarly, BetaAlign can be trained to capture the evolutionary dynamics of different regions within the genome, e.g., introns versus promotors versus other non-coding regions. Moreover, this method has the potential to capture variations among different types of protein-coding genes, e.g., transmembrane versus cytoplasmatic proteins.

One can argue that BetaAlign heavily depends on the simulator. In other words, if the simulator does not reflect biological realism, the resulting aligner may produce inaccurate MSAs for empirical data. However, the design of BetaAlign allows switching between simulators easily. This allows research groups to focus on mimicking the sequence evolutionary process to create better simulators, which in turn, will improve the alignment accuracy. Since tuning a simulator is an easier task than creating a new alignment method, we anticipate a shift in the scientific community from developing better aligners to developing more realistic simulation schemes. BetaAlign can learn those specific alignment rules without the need to specifically provide them as input to the program. This feature abolishes the need to provide any assumptions regarding the evolutionary process at the development stage of the aligner. For example, if a research group assumes that a translocation or a recombination has happened between proteins of different species, they can create a simulator reflecting this process and train BetaAlign to align those sequences, instead of developing new methods to consider translocation or recombination (to the best of our knowledge, current aligners do not consider these types of events).

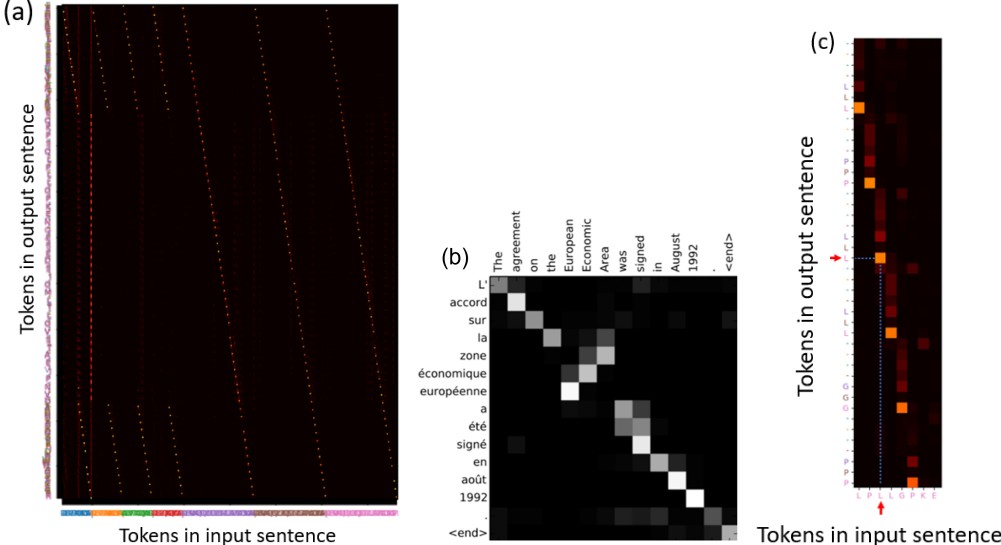

Figure 8: Comparing the attention maps of BetaAlign and other NLP tasks. The x-axis is the input tokens, and the y-axis is the output tokens. The attention splits between all the input tokens when generating the next token. (a) The attention map extracted from the last layer of the BetaAlign's decoder. (b) Attention map extracted from translating an English sentence. The attention map in (b) focuses on one location of the input sentence while BetaAlign focuses on multiple locations, one for each of the unaligned sequences. To visualize this, we colored each of the unaligned sequences in a different color as well as the amino acid that refers to this unaligned sequence. When zooming into the map, there is a step of the attention of the four left diagonals, that we identified as an indel. In this indel, the first "|" has a meaningful part. (c) Zooming into the rightmost diagonal of (a). The red arrows represent the corresponding output and input characters "L".

In this paper, we have demonstrated the power of applying NLP-inspired deep learning methods to sequence alignment. We anticipate that deep learning methods originating in NLP will become more common for solving numerous computation-biology tasks, e.g., to infer ancestral sequences or to detect post-translation modification sites in proteins. Future development will benefit from the establishment of benchmark datasets, developing better pre-training models for biological sequences, and designing dedicated transformers for nucleotide and protein sequences.

## 4.1 LIMITATIONS

BetaAlign has some limitations that impact its effectiveness and utility. First, in its current form it is unable to align very long sequences from hundreds of species. Scaling up the algorithm requires further algorithmic improvements. Second, the application of the methodology requires GPUs for optimal performance, which may not always be available to researchers. Third, BetaAlign heavily relies on specific probabilistic evolutionary models. Evolutionary aspects that are not accounted for in the model will not be captured by the trained algorithm. For example, the models applied assume that the indel dynamics is homogenous across the alignment. However, it is known that indels are more common in the N and C termini of proteins compared to the rest of the sequence. The transformer cannot learn this aspect as it relies on simulations in which homogeneity of the indel process was assumed.

## 5 ACKNOWLEDGEMENTS

Y.B. and T.P. have received funding from the Israel Science Foundation (Grants 448/20 and 2818/21, respectively). Y.B. was supported by an Azrieli Foundation Early Career Faculty Fellowship. E.D., O.A., E.W., N.E., M.A., and G.L. were supported in part by a fellowship from the Edmond J. Safra Center for Bioinformatics at Tel Aviv University.

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

## A  SUPPLEMENTARY INFORMATION

### A.1  SUPPLEMENTARY METHODS

#### A.1.1  GENERATING TRUE MSAS USING SPARTAABC

We used SpartaABC (Loewenthal et al., 2021) to generate millions of true alignments. SpartaABC requires the following input: (1) a rooted phylogenetic tree, which includes a topology and branch lengths; (2) a substitution model (amino acids or nucleotides); (3) root sequence length; (4) the indel model parameters, which include: insertion rate ($R_I$), deletion rate ($R_D$), a parameter for the insertion Zipfian distribution ($A_I$), and a parameter for the deletion Zipfian distribution ($A_D$). MSAs were simulated along random phylogenetic tree topologies generated using the program ETE version 3.0 (Huerta-Cepas et al., 2016) with default parameters.

We describe the data created for Fig. 4 in Section 3.1. We generated 395,000 and 3,000 protein MSAs with ten sequences that were used as training and testing data, respectively. We generated the same number of DNA MSAs. For each random tree, branch lengths were drawn from a uniform distribution in the range $(0.5, 1.0)$. Next, the sequences were generated using SpartaABC with the following parameters: $R_I, R_D \in (0.0, 0.05)$, $A_I, A_D \in (1.01, 2.0)$. The alignment lengths as well as the sequence lengths of the tree leaves vary within and among datasets as they depend on the indel dynamics and the root length. The root length was sampled uniformly in the range $[32, 44]$. Unless stated otherwise, all protein datasets were generated with the WAG+G model, and all DNA datasets were generated with the GTR+G model, with the following parameters: (1) frequencies for the different nucleotides $(0.37, 0.166, 0.307, 0.158)$, in the order "T", "C", "A" and "G"; (2) with the substitutions rate $(0.444, 0.0843, 0.116, 0.107, 0.00027)$, in the order "a", "b", "c", "d", and "e" for the substitution matrix (Fig. 9).

$$Q = \begin{array}{c} \\ \\ T \\ C \\ A \\ G \end{array} \begin{pmatrix} \cdot & a\pi_C & b\pi_A & c\pi_G \\ a\pi_T & \cdot & d\pi_A & e\pi_G \\ b\pi_T & d\pi_C & \cdot & f\pi_G \\ c\pi_T & e\pi_C & f\pi_A & \cdot \end{pmatrix} \begin{array}{c} To \\ \\ T \\ C \\ A \\ G \end{array} \quad From$$

Figure 9: Substitution matrix.

#### A.1.2  GENERATING LONG MSAS

For the dataset used in Fig. 5, we generated 395,000 and 3,000 protein MSAs with five sequences, which were used as training and testing data, respectively. We generated the same number of DNA MSAs. As we used segments in those datasets, we divided the alignments into smaller pieces (see Section 2.2) with a fixed size of 110. Hence, for the protein dataset the number of sentences in the training data was 2,857,364. For the DNA dataset, the number of sentences in the training data was 2,424,287. The number of testing segments varies among datasets as it depends on the transformer results of the previous segment. Of note, when using segments, the number of resulting sentences is higher than the number of alignments (depending on the alignment length and the segment size). In contrast, when aligning shorter sequences, each alignment is a sentence. We used the same SpartaABC parameters as were used for generating the dataset for Fig. 4 (see Section A.1.1), except for a root length sampled uniformly from the interval $[480, 1, 120]$ and a branch length drawn from a uniform distribution in the range $(0.1, 0.15)$.

#### A.1.3  EVALUATING ACCURACY OF INFERRED ALIGNMENTS

Alignment accuracy is often quantified by the column score (CS), which counts the number of alignment columns in the inferred alignment that are identical to the "true" alignment (Penn et al., 2010). We note that we request both the coordinates and the characters of each column to match. For example, consider the case that in a true pairwise alignment, position 17 of the first sequence is an "A", which is matched to an "A" in position 19 of the second sequence. If in the inferred alignment, position 17 of the first sequence matches position 21 in the second sequence, and they are both "A",

this is not considered a match as the coordinates are not the same. To normalize this score to the range of $[0, 1]$, we divided the CS by the total number of columns in the true MSA. The CS-error is defined as $1.0 -$ CS-score, and thus reflects the level of disagreement between the inferred and true MSAs.

### A.1.4  COVERAGE

When the trained BetaAlign is applied to test data, the output is a sequence of tokens. It is possible for this sequence of tokens to not reflect a biologically meaningful alignment. For example, while an alignment algorithm, in general, should only introduce new gap characters, here, our algorithm sometimes wrongly mutates the characters of the input sequence or outputs an alignment in which the sequences are not of the same length (although, by alignment definition, they should be). We call such an output an "invalid alignment". Such cases are illustrated in Fig. 10. A detailed explanation about this metric is at Section A.1.10.

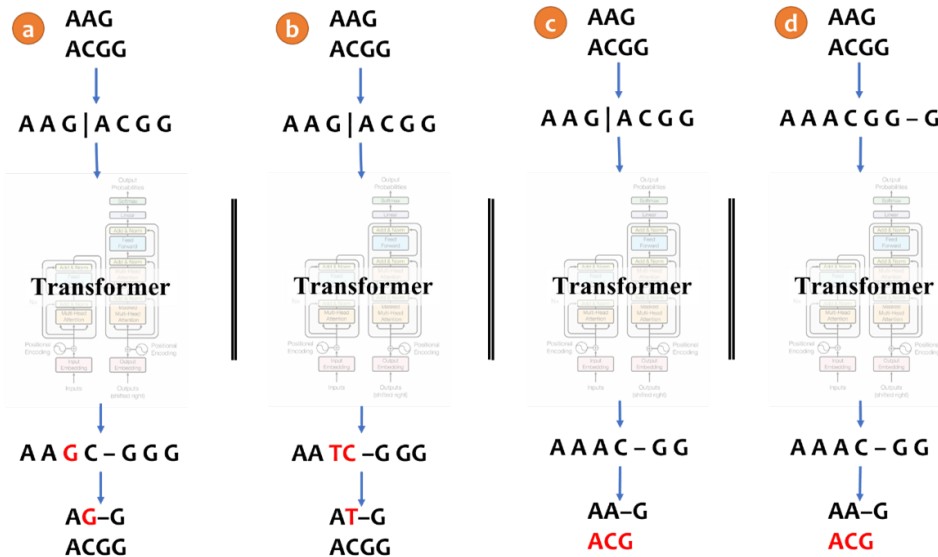

Figure 10: Invalid alignments. (a) Although the first input sequence is "AAG" , the first row in the resulting pairwise alignment is "AG–G", which is invalid as the algorithm not only introduced gaps but also mutated a character. Here, sequences were encoded using the `Concat` representation and decoded with `Spaces`; (b) A similar problem may arise when decoding with `Pairs`; (c) An example of an invalid pairwise alignment when applying the `Concat` and `Spaces` representations, stemming from an unequal length of rows in the resulting MSA. Note that this problem can never appear using `Pairs`; (d) A similar problem may arise in the `Crisscross` encoding. The different representation schemes are explained in Section A.1.8. The transformer architecture illustration is adapted from Vaswani et al. (2017).

It is possible to train several transformers on the same training dataset. These transformers can have different configurations, e.g., as a result of using different optional tunable parameters. One such tunable parameter is, for example, the learning rate. We train a set of transformers on the same training data. It is often the case that one transformer outputs an invalid alignment, while others output a valid alignment (as measured in Section A.2.3). We use only the valid alignments to determine the final output. The performance of BetaAlign is always computed on the output of the original transformer. We term the percentage of cases for which we can obtain valid alignments as "coverage". The more transformers we use, the higher the "coverage", and thus the higher our confidence in the final output. Of note, using alternative transformers also provides alternative alignments, which can be used, for example, to estimate alignment uncertainty (Wu et al., 2012; Sela et al., 2015).

### A.1.5  Preprocessing the Input for the Machine Translation Task

To train the NLP models, we first represent the input unaligned sequences in one of the two source languages, as explained in Section A.1.8 (`Concat` or `Crisscross`). The true MSAs were also represented in the target languages (`Pairs` or `Spaces`). The actual training of the NLP transformers as well as the pre-processing step were done using the Fairseq library (Ott et al., 2019), which is implemented in Python. Six text files are used as input for the pre-processing step before actually running the transformer: unaligned sequences and true MSAs, for training, validation, and testing data.

### A.1.6  Preprocessing Inputs and Outputs

In addition, for running the preprocessing steps, one has to declare names for the input and output languages. Moreover, one can explicitly state if the same dictionary is used for both the source and target languages, i.e., whether a joint dictionary should be assumed. In our case, when we apply the combinations of `Concat` and `Spaces` or `Crisscross` and `Spaces`, the dictionary is joint, while for the combination `Concat` and `Pairs`, the dictionary is not joint and hence two different dictionaries are internally generated by the library (one for the source language and the other for the target language). The output of this preprocessing step is binary files that are later used as input to the transformer. This preprocess step allows running different configurations of the transformers (or potentially, other machine-learning models) on the same input files, without the need to perform the same computations on the input files over and over again.

### A.1.7  Training the Transformers

We first assessed various transformer configurations, which differ in their training parameters: max tokens, learning rate and warmup updates. Based on their performance on the pairwise nucleotide and protein alignment datasets (Supplementary Table 2), we selected two optimal transformer configurations, which we call "original" and "alternative". The learning rate and warmup values for both transformers are 5E-5 and 3,000, respectively. The max token parameter values are 4,096 and 2,048 for the original and alternative transformers, respectively.

We used label-smoothed cross entropy (Szegedy et al., 2015) to compute the loss of the model with a dropout (Srivastava et al., 2014) rate of 0.3. We used "Adam" (Kingma & Ba, 2014) as the optimizer of the model with 0.9 forgetting factors for gradients and 0.98 for the and second moments of gradients.

All model evaluations are executed on GPU machines, Tesla V100-SXM2-32GB. The Weights and Biases platform (https://wandb.ai) was used to follow the progress of the training steps and helped decide when to stop the learning processes.

### A.1.8  Comparing Different Representations

We examined several source and target languages for the sequence-to-sequence learning task. In addition to `Concat` (see Section 1) we consider another possibility for encoding the input sequences, and we call the corresponding language `Crisscross`. In this language, the first token is the first character of the first sequence and the second token is the first character of the second sequence. From the perspective of a transformer, it helps because it needs to learn closer rather than longer dependencies, compared to `Concat`. As the input sequences are not necessarily of the same length, a gap character ("–") is placed if no more characters are available for one of the sequences. For example, for the example input sequences in Fig. 1 ("AAG" and "ACGG"), the input sentence would be "A A A C G G - G". Thus, the number of tokens is always twice the length of the longest sequence.

In addition to `Spaces`, we consider an alternative decoding for the output alignments, with the corresponding language called `Pairs`. We note that each column of the output alignment can be one of the following: "AA", "AC", ..., "A-", "CA", ..., "T-", "-A", ..., "-T". In `Pairs`, each such possibility is a token. Thus, for nucleotide sequences, the dictionary size is 24. For protein sequences, the dictionary size is 440. Note that "- -" (gap and another gap) is an illegal token.

We encoded the same datasets with different representations: (1) `Concat` and `Pairs`; (2) `Concat` and `Spaces` (see Section 1); (3) `Crisscross` and `Spaces`. Then, we trained two transformers on each of the datasets, e.g., two transformers on the `Concat` and `Pairs`, and two different transformers on `Concat` and `Spaces`. The transformers were trained with the same tunable parameters. We compared the transformers' performance on the same test data. This comparison was done on dataset PD2 (see Supplementary Table 2). Note, that we did not use the `Pairs` representations when shifting from pairwise to multiple alignment because the dictionary of pairwise and multiple alignments are different, which disenables transfer learning.

### A.1.9 TRANSFER LEARNING IMPLEMENTATION

In our work, transfer learning was repeatedly used for training the transformers. The first protein transformer was trained on a simple dataset of pairwise amino acid sequences (we denote this dataset "PD1", for protein dataset 1). Its weights were randomly sampled with default values of the Fairseq library. The resulting trained transformer is termed "PT1", for protein transformer 1. PT1 was next trained on PD2, resulting in PT2, etc. A similar process was used to train the nucleotide-based transformers (NT1, NT2, etc.) on nucleotide (DNA) datasets (ND1, ND2, etc.) Of note, transfer learning was used in this study only when the previous and the currently processed data are encoded using the same representations, i.e., they share the same dictionaries.

### A.1.10 CALCULATING COVERAGE

Each alignment was passed through both the original and alternative transformers. A valid alignment is defined by the following rules: (1) All alignment rows are of the same length; (2) When removing all gaps from each row of the alignment, the result equals to the unaligned input sequences. For example, for the unaligned inputs: "AAG" and "ACGG", the inferred alignment, "A-AG" over "ACGG" is valid, whereas for the same input the alignment "A-AG" over "ACGC" is invalid (see Fig. 10). The coverage is defined as the fraction of valid alignments.

### A.1.11 GENERATING MISSPECIFICATION DATASET

To create misspecification datasets, we simulated alignments that have four different regions, each evolving under a different indel model, i.e., each part of the alignment has different indel parameters. The $R_I$ and $R_D$ are sampled from the ranges: $[0.01, 0.02]$, $[0.02, 0.03]$, $[0.03, 0.04]$, $[0.04, 0.05]$ for the first, second, third and fourth regions of the alignment, respectively. The $A_I$ and $A_D$ are sampled from the ranges: $[1.01, 1.2]$, $[1.2, 1.4]$, $[1.4, 1.6]$, $[1.6, 1.8]$ for the different regions, respectively. Those values were assumed both when simulating nucleotide and protein datasets. For protein sequences, each region was simulated assuming a different amino-acid replacement matrix: JTT (Jones et al., 1992), BLOSUM (Henikoff & Henikoff, 1992), mtART (Abascal et al., 2007), and LG (Le & Gascuel, 2008), for the first to the fourth domains, respectively. For nucleotide sequences, the GTR1, JC, GTR2, JC, were assumed for each region, respectively (Pupko & Mayrose, 2020). The GTR1 and GTR2 frequencies are (0.37, 0.166, 0.307, 0.158) and (0.265, 0.182, 0.171, 0.382) for the frequency of "T", "C", "A", and "G", respectively. The assume entries in the rate matrix were: (0.444, 0.0843, 0.116, 0.107, 0.00027) and (1.387, 1.79, 0.431, 0.321, 4.947), for "a", "b", "c", "d", and "e" as stated in matrix Q (see Section A.1.1), for GTR1 and GTR2, respectively.

### A.1.12 EXTRACTING THE ATTENTION MAPS

In the Fairseq library, which we used for training the transformers, there is an option to extract the attention map. Since using this option resulted in a bug, we fixed it by changing the source code of the library. Once the issue was fixed, we could save the attention maps.

### A.1.13 COMPARING AGAINST OTHER ALIGNMENT PROGRAMS

We compared BetaAlign against the following programs, used with default parameters: MUSCLE v3.8.1551 (Edgar, 2004), MAFFT v7.475 (Katoh & Standley, 2013), PRANK v.150803 (Löytynoja & Goldman, 2008), T-Coffee Version.12.00.7fb08c2 (Notredame et al., 2000), ClustalW 2.1 (Larkin et al., 2007), and DIALIGN dialign2-2 (Morgenstern, 2004).

## A.2 SUPPLEMENTARY DISCUSSION

### A.2.1 CHARACTERIZING THE ERROR OF DIFFERENT ALIGNERS

To better understand the limitations of the various multiple sequence aligners, we analyzed the resulting alignments with respect to 25 features, e.g., the resulting alignment length, the total number of gaps, and the average gap size. We then compared these features to the true alignment. Fig. 11 shows the level the resulting alignment distorts these features as a heatmap. The closest the value to zero, the less biased the aligner is with regard to this feature. For example, a deviation of $-0.09$ in alignment length means that the aligner tends to produce alignments that are $9\%$ shorter than the true alignment (over-aligning). BetaAlign's deviations range from $-6\%$ to $6\%$ and are lower than other aligners. Other aligners are especially biased with respect to long indels. We hypothesize that this is due to their implicit assumptions regarding the distribution of the indels, which is different from the distributions used to simulate these data.

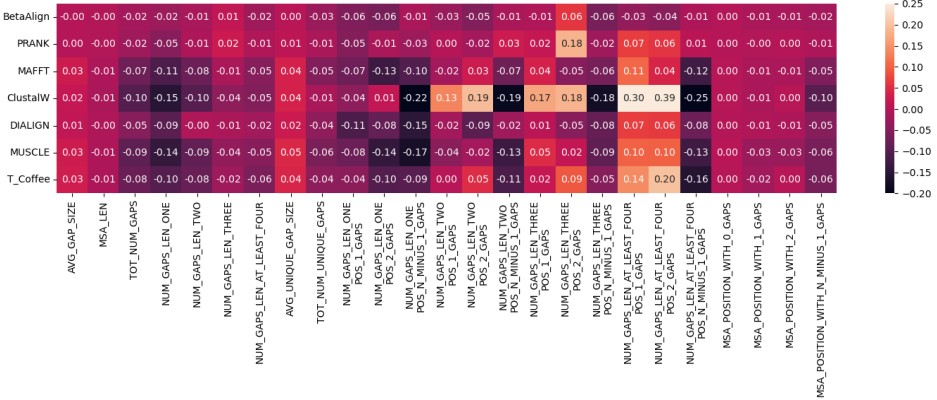

Figure 11: Level of alignment attributes distorted by different aligners. For each feature and for each alignment, shown is the percent deviation comparing the resulting alignment to the true simulated alignment. The data used in this section are SPD12. For a detailed description of the features used, see Loewenthal et al. (2021).

### A.2.2 TESTING ALTERNATIVE REPRESENTATIONS SCHEMES

Supplementary Table 1 displays the performance of transformers trained with various encodings for pairwise protein alignment. All six variants had similar accuracy, as measured by the CS-score. The `Pairs` target language resulted in the highest coverage. For the task of multiple sequence alignments, we decided not to work with the generalization of the `Pairs` target language because in this case the size of the dictionary increases exponentially with the number of input sequences (if we denote $n$ as the number of input sequences, the output protein dictionary size is $22^n - 1$). Furthermore, because the dictionary size depends on $n$, the `Pairs` target language hinders using transfer learning. Note that the `Spaces` and `Crisscross` dictionaries do not change when increasing the number of input sequences. As the performance using the `Concat` source language was slightly better than that of the `Crisscross` source language, the former was applied for all analyses.

### A.2.3 COVERAGE RESULTS

When generating the results of Fig. 4 in the main text, a small fraction of the resulting alignment was invalid (see Section A.1.4). Changing the tunable parameters of the transformers led to different deep-learning models. We harnessed this feature of the neural network models and generated an additional transformer for aligning nucleotide sequences as well as an additional transformer for aligning protein sequences (we call each of these transformers "alternative"). When we aligned the invalid cases with the alternative transformers, a large fraction yielded valid alignments. Thus, every time the "original" transformer failed, the results of the alternative transformer were considered. This methodology substantially increased the coverage, i.e., the percentage of MSAs that were suc-

Table 1: Transformers performance trained on pairwise protein alignments with source and target languages (dataset PD2). Each combination of source and target language was tested with two values for the max-token parameter, resulting in two transformers: "original" and "alternative".

| TRANSFORMER NAME | SOURCE LANGUAGE | TARGET LANGUAGE | MAX TOKENS | CS-SCORE | COVERAGE |
|---|---|---|---|---|---|
| original | Concat | Pairs | 4096 | 0.997 | 0.939 |
| alternative | Concat | Pairs | 2048 | 0.997 | 0.952 |
| original | Concat | Spaces | 4096 | 0.997 | 0.835 |
| alternative | Concat | Spaces | 2048 | 0.998 | 0.635 |
| original | Crisscross | Spaces | 4096 | 0.991 | 0.503 |
| alternative | Crisscross | Spaces | 2048 | 0.995 | 0.636 |

cessfully aligned. Specifically, the fraction of datasets that could not be aligned was less than 2% both for nucleotide and protein sequences (Fig. 12).

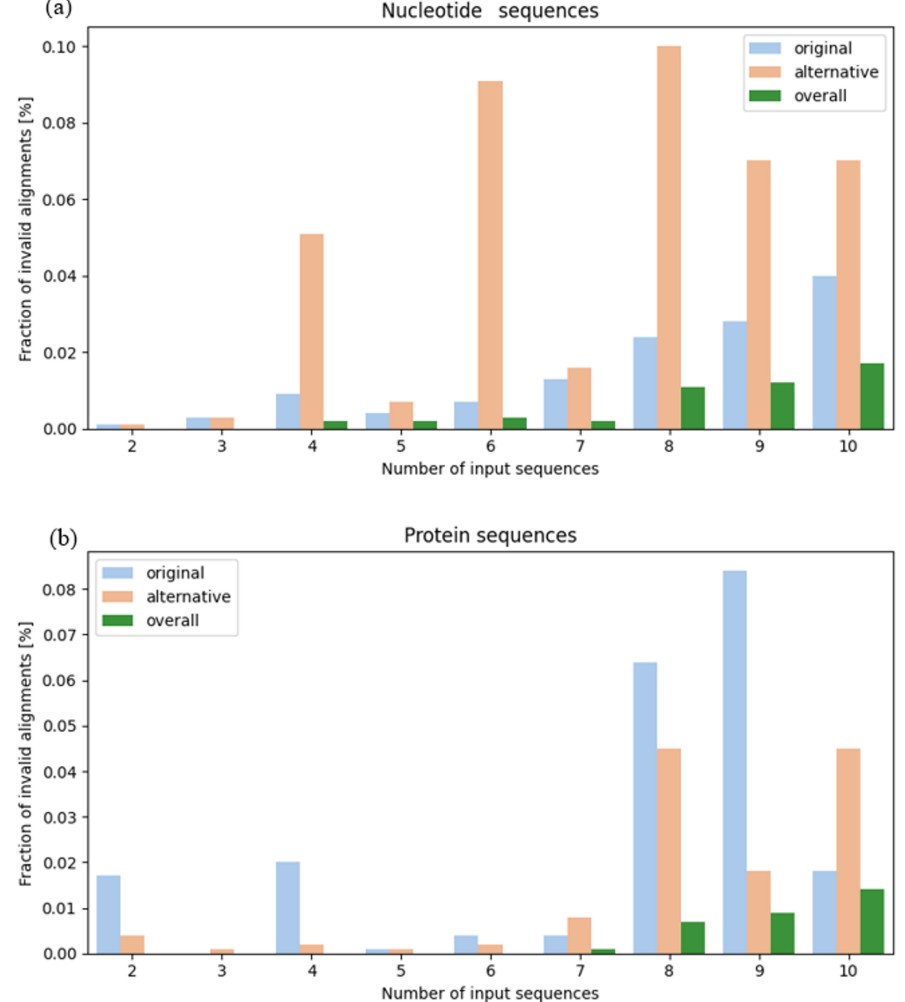

Figure 12: The fraction of invalid alignments for nucleotides (a) and protein (b) sequences as a function of the number of inputs sequences. For each dataset, two different transformers were trained. The fraction of invalid alignments of the "original" and "alternative" transformers are in blue and orange, respectively. In green is the fraction of invalid MSAs shared by both transformers.

Table 2: The generation of datasets used in this work. The $R_I \& R_D$ and the $A_I \& A_D$ refer to the indel parameters of the SpartaABC. The root length was sample uniformly from in the range (minimum sequence length times 0.8, maximum sequence length times 1.1). Datasets names reflect to the order in which the transformers were trained. For example, the first protein transformer was trained on dataset PD1, then the optimized weights were the starting point of PD2, etc. Similarly, the nucleotides transformer was first trained on ND1. Tables (a), (b) refer to nucleotide and protein datasets, respectively. Table (c) refers to special datasets (hence the "S" at the start of the dataset name). In this table, datasets SND1 and SPD1LG are the datasets used to create Fig. 4. Of note, dataset SPD1LG was generated with the LG amino-acid replacement matrix. The other datasets, SND2 to SND5 and SPD2 to SPD9 refer to nucleotide and protein sequences used for the "segmented transformers", respectively. SND6 and SPD10 refers to the misclassification datasets, and thus, the indel parameters were sampled separately for each "region" (see Section A.1.11).

| DATASET | BRANCH LENGTH | ROOT LENGTH | $R_I \& R_D$ | $A_I \& A_D$ | INPUT SEQUENCES NUMBER |
|---|---|---|---|---|---|
| ND1 | 0.03 - 0.1 | 50 - 60 | 0.0 - 0.05 | 1.01 - 2.0 | 2 |
| ND2 | 0.03 - 0.3 | 100 - 300 | 0.0 - 0.05 | 1.01 - 2.0 | 2 |
| ND3 | 0.3 - 0.6 | 200 - 300 | 0.04 - 0.05 | 1.01 - 2.0 | 2 |
| ND4 | 0.1 - 0.3 | 50 - 60 | 0.04 - 0.05 | 1.01 - 2.0 | 3 |
| ND5 | 0.15 | 55 | 0.5 | 1.0 - 1.01 | 3 |
| ND6 | 0.15 | 55 | 0.5 | 1.01 | 3 |
| ND7 | 0.15 | 55 | 0.5 | 1.5 | 3 |
| ND8 | 0.05 - 0.1 | 55 | 0.0 - 0.05 | $1.01 - 2$ | 4 |
| ND9 | 0.05 - 0.1 | 55 | 0.03 - 0.05 | $1.01 - 2$ | 4 |
| ND10 | 0.07 - 0.1 | 35 - 45 | 0.0 - 0.05 | $1.01 - 2$ | 5 |
| ND11 | 0.08 - 0.09 | 37 - 42 | 0.03 - 0.05 | $1.01 - 2$ | 5 |
| ND12 | 0.09 | 40 | 0.04 | 1.3 | 5 |
| ND13 | 0.05 - 0.1 | 55 | 0.02 - 0.03 | 1.0 - 1.1 | 4 |
| ND14 | 0.9 | 40 | 0.01 - 0.02 | 1.35 - 1.45 | 5 |
| ND15 | 0.07 - 0.1 | 35 - 45 | 0.0 - 0.05 | $1.01 - 2$ | 7 |
| ND16 | 0.07 - 0.1 | 70 - 80 | 0.0 - 0.05 | $1.01 - 2$ | 7 |

| DATASET | BRANCH LENGTH | ROOT LENGTH | $R_I \& R_D$ | $A_I \& A_D$ | INPUT SEQUENCES NUMBER |
|---|---|---|---|---|---|
| PD1 | 0.03 - 0.05 | 30 - 40 | 0.0 - 0.05 | 1.01 - 2.0 | 2 |
| PD2 | 0.05 - 0.1 | 70 - 80 | 0.04 - 0.05 | 1.01 - 2.0 | 2 |
| PD3 | 0.1 - 0.3 | 200 - 250 | 0.04 - 0.05 | 1.01 - 2.0 | 2 |
| PD4 | 0.03 - 0.1 | 30 - 40 | 0.0 - 0.05 | 1.01 - 2.0 | 3 |
| PD5 | 0.1 - 0.2 | 50 - 60 | 0.04 - 0.05 | 1.01 - 2.0 | 3 |
| PD6 | 0.15 | 50 | 0.05 | 1.01 | 3 |
| PD7 | 0.15 | 50 | 0.05 | 1.5 | 3 |
| PD8 | 0.05 - 0.1 | 30 - 40 | 0.0 - 0.05 | 1.01 - 2 | 4 |
| PD9 | 0.075 | 35 | 0.03 | 1.07 | 4 |
| PD10 | 0.04 - 0.08 | 30 - 40 | 0.0 - 0.05 | 1.01 - 2 | 5 |
| PD11 | 0.04 - 0.08 | 30 - 40 | 0.0 - 0.05 | 1.01 - 2 | 6 |
| PD12 | 0.1 | 30 - 40 | 0.0 - 0.05 | 1.01 - 2 | 6 |
| PD13 | 0.1 | 30 - 40 | 0.03 - 0.05 | 1.01 - 2 | 6 |
| PD14 | 0.07 - 0.1 | $25 - 35$ | 0.0 - 0.05 | 1.01 - 2 | 7 |
| PD15 | 0.08 - 0.09 | 27 - 32 | 0.03 - 0.05 | 1.01 - 2 | 7 |
| PD16 | 0.09 | 30 | 0.04 | 1.3 | 7 |
| PD17 | 0.05 - 0.1 | 40 | 0.0 - 0.05 | 1.01 - 2 | 10 |
| PD18 | 0.07 - 0.1 | 25 - 35 | 0.04 - 0.05 | 1.01 - 2 | 7 |

| DATASET | BRANCH LENGTH | ROOT LENGTH | $R_I \& R_D$ | $A_I \& A_D$ | INPUT SEQUENCES NUMBER |
|---|---|---|---|---|---|
| SND1 | 0.05 - 0.1 | 40 | 0.0 - 0.05 | 1.01 - 2.0 | 10 |
| SND2 | 0.1 - 0.15 | 600 - 800 | 0.0 - 0.05 | 1.01 - 2 | 3 |
| SND3 | 0.1 - 0.15 | 600 - 800 | 0.0 - 0.05 | 1.01 - 2 | 4 |
| SND4 | 0.1 - 0.15 | 100 - 120 | 0.0 - 0.05 | 1.01 - 2 | 5 |
| SND5 | 0.1 - 0.15 | 600 - 800 | 0.0 - 0.05 | 1.01 - 2 | 5 |
| SND6 | 0.05 - 0.1 | 40 | Dynamic | Dynamic | 10 |
| SPD1 | 0.05 - 0.1 | 40 | 0.0 - 0.05 | 1.01 - 2.0 | 10 |
| SPD1LG | 0.05 - 0.1 | 40 | 0.0 - 0.05 | 1.01 - 2.0 | 10 |
| SPD2 | 0.08 - 0.15 | 220 - 280 | 0.0 - 0.05 | 1.01 - 2 | 3 |
| SPD3 | 0.08 - 0.15 | 50 - 120 | 0.0 - 0.05 | 1.01 - 2 | 4 |
| SPD4 | 0.08 - 0.15 | 250 - 350 | 0.0 - 0.05 | 1.01 - 2 | 4 |
| SPD5 | 0.08 - 0.2 | 600 - 800 | 0.0 - 0.05 | 1.01 - 2 | 4 |
| SPD6 | 0.08 - 0.2 | 600 - 800 | 0.0 - 0.05 | 1.01 - 2 | 7 |
| SPD7 | 0.08 - 0.2 | 600 - 800 | 0.0 - 0.05 | 1.01 - 2 | 5 |
| SPD8 | 0.1 - 0.15 | 600 - 800 | 0.0 - 0.05 | 1.01 - 2 | 4 |
| SPD9 | 0.1 - 0.15 | 600 - 800 | 0.0 - 0.05 | 1.01 - 2 | 5 |
| SPD10 | 0.05 - 0.1 | 40 | Dynamic | Dynamic | 10 |
| SPD11 | 0.07 - 0.12 | 40 | $R_I$: 0.01269 - 0.0155 $R_D$: 0.035 - 0.043 | $A_I$: 1.512 - 1.85 $A_D$: 1.423 - 1.739 | 7 |
| SPD12 | 0.07 - 0.12 | 40 | Dynamic | Dynamic | 7 |
| SPD13 | 0.07 - 0.12 | 50 | Dynamic | Dynamic | 7 |

### A.2.4 ADDITIONAL DATASETS

The parameters that used to generate the different datasets are listed in Supplementary Table 2.

### A.2.5 DATA FOR SPECIFIC FIGURES

To create Fig. 4, and Fig. 12 we used the same datasets, SND1 and SND1LG for the nucleotide and protein dataset, respectively. The datasets for Fig. 5 are SND5 and SPD9 for the nucleotide and protein dataset, respectively. We have used SPD11, SPD12 and SPD13 to create Fig. 6. In Fig. 7, the transformers that we used were trained on datasets SND1 and SPD1. The test datasets that include spatial variation are SND6 and SPD10 for nucleotide and protein datasets, respectively. Shown in Fig. 8a and Fig. 8c are attention maps of different resolutions extracted from one of the protein alignments composed of seven sequences (dataset PD14). Dataset SPD12 was used to create Fig. 11.

