# OpenReview forum: "Multiple sequence alignment as a sequence-to-sequence learning problem"
_ICLR.cc/2023/Conference — ICLR 2023 poster_

### Official Review · Reviewer_8ZqM · 2022-10-24

**Confidence:** 4
**Correctness:** 3
**Technical Novelty And Significance:** 3
**Empirical Novelty And Significance:** Not applicable
**Recommendation:** 6

**Clarity, Quality, Novelty And Reproducibility:**

Clarity: The paper is generally clear but is missing some key elements such as a Related Works section (some of it is relegated to the Appendix). Additional qualification of the settings in which this method would work well vs poorly as well as additional humility("This pioneering paper") would make extremely clear what problem this alignment method is tackling. For example, this method does not seem well-suited to aligning a large number of sequences (or at the least it wasn't tested) or aligning a large number of sets of sequences. The compute cost can be prohibitive in some settings and the splitting mechanism is not general to any form of multiple sequence alignment and qualifying the exact type of problem this is trying to tackle in the introduction and abstract would likely be greatly appreciated by the reader.

Quality: The paper develops a good idea and executes it over a set of thorough experiments from a ML perspective. However, a few additional experiments are likely necessary to demonstrate its qualities as a viable piece of the bioinformatic workflow.

Novelty: The paper adopts some interesting ideas from NLP and likelihood-free inference and applies it to the MSA problem.

Reproducibility: In order for this paper to be accepted, code should be released given the nature of the problem that is being tackled and its place in the bioinformatics workflow.

**Strength And Weaknesses:**

Strengths:
- The paper likely improves the SOTA for some applications of multiple sequence alignments.
- The paper develops an interesting splitting strategy to apply this method for longer sequences.
- The leveraging of simulators allows the method to more closely tailor the alignment to the phylogenetic process.

Weaknesses:
- While model misspecification is addressed in simulator parameter space, it isn't addressed in the mismatch between simulators and real data. An experiment that applies to the "gold standard" real alignment data while the model is trained on simulator data would be a vital experiment to convince the reader that the effects won't decay in the sim2real setting as often occurs in genomics, robotics, and vision.
- Compute can be prohibitive(only applied to a moderate number of sequences) even with the splitting strategy as the number of sequences increases or the number of alignments that need to be performed. Further analysis of compute time(including training) is likely one of the crucial pragmatic aspects of a bioinformatic piece of software that is noticeably missing.
- The splitting strategy likely will not work multiple sequence local alignments given how to split the sequences will likely be unclear.
- More discussion around inferring the correct simulator from data would greatly improve the method's viability.

Minor Comment:
- Providing a more substantial Related Work section in the Main Text would improve its readability and context for this paper.
- Connection to the likelihood-free inference literature for related works (Beaumont et al 2002[1] where ABC was presented and Chan et al 2018 [2] which combines simulators + deep NNs + sets of sequences).

[1] Beaumont et al (2002). Approximate Bayesian Computation in Population Genetics.
[2] Chan et al (2018). A Likelihood-Free Inference Framework for Population Genetic Data using Exchangeable Neural Networks.

**Summary Of The Paper:**

The paper proposes using sequence-to-sequence translation from NLP as the core of a Multiple Sequence Alignment. They train their model on phylogenetic simulators and compare against standard alignment methods.

**Summary Of The Review:**

The paper provides an interesting combination of ideas and applies them to the MSA problem. On technical and empirical merit, this is interesting and a good paper. The key weaknesses for the methods usability are somewhat uncovered primarily (1) extension to real data (2)  addressing of compute challenges and (3) how to infer the correct simulator parameters. Furthermore, the exposition is a bit grandiose that sets the reader up for high expectations that detracts from the overall work. More humility in word choice, providing reader context of related work, and defining clearly the set of MSA tasks the method excels in and struggles at would greatly improve the paper.

---

> ### Author Response · Authors · 2022-11-18
> **Author Response to 8ZqM [2 / 2]**
>
> *Comment*: “The splitting strategy likely will not work multiple sequence local alignments given how to split the sequences will likely be unclear.”
>
>
> We have not tested local alignments. We hypothesize that in those kinds of scenarios the value of “segment size” is important. It should be higher than max gap size (and in the local alignment, should be higher than the gaps at both ends) in order to successfully align the sequences.
>
>
> *Comment*: “While model misspecification is addressed in simulator parameter space, it isn't addressed in the mismatch between simulators and real data. An experiment that applies to the "gold standard" real alignment data while the model is trained on simulator data would be a vital experiment to convince the reader that the effects won't decay in the sim2real setting as often occurs in genomics, robotics, and vision.”
>
> We are certain that our simulator isn’t representative of real biological data. There are complex evolutionary events that are not considered. However, there are issues regarding evaluation on what is considered the “gold standard” in molecular evolution [2], [3]. To the best of our knowledge the simulator used is the only one which considers different evolution distributions for insertion and deletion. We are planning to improve our simulator to better capture the evolution.
>
>
> *Comment*: “Further analysis of compute time(including training) is likely one of the crucial pragmatic aspects of a bioinformatic piece of software that is noticeably missing.”
>
> The training time vastly differs from the starting point of the transformer and the dataset trained on. Training on a new dataset with the same number of sequences (e.g., training the Drosophila dataset after training for the 7 sequences dataset) took several hours on a single GPU. The inference time is a few seconds on a single CPU for short sequences and could rise to minutes. Of course, the inference time will be shortened on a GPU.
>
>
> *Comment*: “More discussion around inferring the correct simulator from data would greatly improve the method's viability.”
>
>
> The approach we present depends on the ability to simulate sequences using a model that reflects the patterns of substitution and indels found in the analyzed empirical alignments. For substitution, the pattern is often inferred using model-selection criteria, as implemented in programs such as ModelTest [4]. SPARA-ABC was recently developed for inferring the indel evolutionary dynamics. This is a Bayesian approach that can provide an accurate estimate of indel dynamics [5]. The model parameters include the rate of insertions and the rate of deletions, relative to substitutions, as well as the shapes of the indel-length distributions (insertions and deletions are allowed to have different length distributions). We expect that such approximate Bayesian computation (ABC) approaches would allow inferring more complicated indel models in the future.
>
>
> If you have any more comments, please let us know.
>
>
>
> [1] - Talia Kustin, Adi Stern, Biased Mutation and Selection in RNA Viruses, Molecular Biology and Evolution (2021).
>
> [2] - David A. Morrison, Why Would Phylogeneticists Ignore Computerized Sequence Alignment?, Systematic Biology, (2009).
>
> [3] -  Iantorno S. Gori K. Goldman N. Gil M. Dessimoz C. Who watches the watchmen? An appraisal of benchmarks for multiple sequence alignment Methods Mol. Biol. (2014).
>
> [4] - Posada D, Crandall KA. MODELTEST: testing the model of DNA substitution. Bioinformatics. (1998).
>
> [5] - G. Loewenthal, D. Rapoport, O. Avram, A. Moshe, E. Wygoda, A. Itzkovitch, O. Israeli, D. Azouri, R. A. Cartwright, I. Mayrose, T. Pupko, A Probabilistic Model for Indel Evolution: Differentiating Insertions from Deletions. Mol. Biol. Evol. (2021).

---

> ### Author Response · Authors · 2022-11-18
> **Author Response to 8ZqM [1 / 2]**
>
> Dear Reviewer,
>
> First and foremost, we are grateful for receiving your detailed feedback it helped us improve our paper.
>
>
> *Comment*: “Providing a more substantial Related Work section in the Main Text would improve its readability and context for this paper.”
>
>
> We added the “related work” section to the text.
>
> Our approach is drastically different from previous concepts of aligners. We don’t consider any scoring scheme but learn it straight from the data. This allows us to consider different types of indel distributions and different rates for insertion and deletion. In addition, we are directly infer the multiple sequence alignment based on all of the sequences instead of using pairs. This allows the transformer to have the entire data at the start of the alignment process.
>
>
> *Comment*: “Additional qualification of the settings in which this method would work well vs poorly…”
>
>
> This method will work best given biological knowledge regarding the evolutionary process. Consider a lab where the accuracy of the alignment is crucial. This lab works with a specific species and understands that only certain parts of a protein are likely to have indels and others that do not (e.g., a conserved domain). Once simulating sequences based on those assumptions, the lab could benefit from an aligner that considers it. Another example could be aligning viruses where it was shown that different substitutions are more likely to happen than others [1]. Fitting the current aligners specifically for this new biological knowledge won’t be as easy as simulating it. We have not tested BetaAlign on hundreds of sequences, but we hypothesize it will be best used for several sequences.
>
>
> *Comment*: “The leveraging of simulators allows the method to more closely tailor the alignment to the phylogenetic process.”
>
>
> We thank the reviewer for this comment. Simulating the evolution process creates the data needed to learn a function between the unaligned sequences and the aligned sequences. This function (i.e., the transformer) can generalize better to complex evolutions events compared to today’s aligners. This is the core of our paper and the novelty it brings.
>
>
> *Comment*: “...additional humility("This pioneering paper")...”
>
>
> We have toned down our language
>
> Since the development of the Needleman–Wunsch algorithm (back in the 1970s), most of the popular aligners are based on dynamic programming. Each of the aligners compared to in this paper is a different variation of it. To the best of our knowledge this is the first time that a learning mechanism is proposed to fully align multiple sequences and is comparable to popular sequence aligners. We hypothesize that this paper could be the basis of the next generation aligners, as our assumptions are more flexible and not limited to the implemented algorithms (e.g., dynamic programming).

---

### Official Review · Reviewer_AUpf · 2022-10-24

**Confidence:** 5
**Correctness:** 2
**Technical Novelty And Significance:** 2
**Empirical Novelty And Significance:** 2
**Recommendation:** 3

**Clarity, Quality, Novelty And Reproducibility:**

The idea is kind of interesting. However, the experimental design and results contain flaws. A main category of methods is missed. And the comparison is unfair.

**Strength And Weaknesses:**

## Major concern

1. Due to the nature of the transformer, there is no guarantee that all input(e.g. ATCG) will be mapped to the output. The MSA will be meaningless if BetaAlign cannot guarantee that input tokens are not corrupted in the resulting alignment.
2. The experiments are conducted on simulated datasets with few sequences in each MSA. It seems that BetaAlign is more time-consuming than traditional methods since each segment needs to pass the transformer once. The alignment comparison between BetaAlign and traditional ones shows that it is comparable, but it remains a question that why we need deep learning here to achieve similar results but consume more time.
3. The authors should mention HMM-based methods[1] as well as other ones incorporating profiles.
4. Datasets based on structural alignment should be used as the benchmark dataset, like BAliBASE. Using simulated data similar to the training set is not fair for traditional ones.

## Minor comments

1. The construction of the training & testing set is ambiguous. Also, the parameters of traditional methods for benchmarking should be stated.
2. The evaluating metric **SC-error** is confusing. Why not use the original CS (column score) or SPS (Sum-of-pairs score)[2]? I think CS is already normalized.
3. Latest downstream tasks of MSA like AlphaFold2 for 3D structure prediction use MSA with around one hundred sequences for promising performance. Figure 4 should show the comparison of a more diverse number of input sequences.

[1]  S. R. Eddy, Accelerated profile HMM searches. *PLoS Comp. Biol.*, 7:e1002195, 2011.

[2] Julie D. Thompson, Frédéric Plewniak, Olivier Poch, A comprehensive comparison of multiple sequence alignment programs, *Nucleic Acids Research*, Volume 27, Issue 13, 1 July 1999

**Summary Of The Paper:**

This paper proposes to treat multiple sequence alignment task as a machine translation task by processing 'concats' into 'spaces'. It is an interesting idea, and the authors resolve the main obstacle of transformer sequence length limitation with segments.

**Summary Of The Review:**

The idea is kind of interesting. However, the motivation is not very clear. Also, the comparison between the proposed method and the traditional MSA methods is not fair. HMM-based methods are missing. The advancement of the method compared to the previous methods is unclear to me.

---

> ### Author Response · Authors · 2022-11-18
> **Author Response to AUpf [2 / 2]**
>
> *Comment*: “Using simulated data similar to the training set is not fair for traditional ones”
>
> This is a critical part of the work, thank you for highlighting this issue. This comment introduces two questions, how could we use biological knowledge to improve current aligners? Could we use aligners with specific assumptions to align data that behave differently? Regarding the first question, there is usually a gap between the hyperparameters of the current aligners and the biological knowledge you know. Tunning those parameters does not scale well when new information about evolution is learned. Current aligners are not as flexible as BetaAlign. Regarding the second question, many researchers are using those aligners for biological data which are much more complex than the simulated data we generated…
>
> Of note, the simulated data provided follow the latest advances in molecular evolution. It assumes a different rate for insertion and deletion and a Zipf distribution for the length of the insertion and deletion ([5], [6], [7], [8]). The results in figures 4 and 5 are of transformers trained on the entire parameter space, meaning that no pre-assumptions were considered. However, we did test the robustness of the different aligners (section “Model Misspecification”) and verified it on the misspecification datasets (where the trained and test data were generated with different distributions).
>
>
> *Comment*: “HMM-based methods are missing”
>
>
> Following the reveiwer’s comment, we tried comparing the various aligners to HMM-based
> methods, specifically HMMER3 [11]. However, HMMER3 requires an input alignment to build
> profiles. Using a profile, it can align additional sequences. The seed alignment is computed by methods such as MAFFT and PRANK. In this program there is no direct way to provide a set of unaligned sequences as input and obtain as output an MSA. Thus, we do not compare to these methods. Of note, previous papers developing aligners also did not compare to HMM-based methods, probably for the same reason [3], [4], [9], [10].
>
>
> *Comment*: “Latest downstream tasks of MSA like AlphaFold2 for 3D structure prediction use MSA with around one hundred sequences for promising performance. Figure 4 should show the comparison of a more diverse number of input sequences”
>
>
> Aligning massive amounts of sequences requires different types of aligners. AlphaFold uses MMseq2 [12] which is mostly used to search against databases. The tested software (and BetaAlign) have limits on the number of input sequences. Those are two different types of software, one optimizing the accuracy and the other type optimizing running time (and hence could align more sequences). It is not common to compare between the two types see previous paper [3], [4], [9], [10] and the one of MMseq2 [12].
>
>
> If you have any more comments, please let us know.
>
>
> [1] - David A. Morrison, Why Would Phylogeneticists Ignore Computerized Sequence Alignment?, Systematic Biology, (2009).
>
> [2] -  Iantorno S. Gori K. Goldman N. Gil M. Dessimoz C. Who watches the watchmen? An appraisal of benchmarks for multiple sequence alignment Methods Mol. Biol. (2014).
>
> [3] - A. Löytynoja, N. Goldman, Phylogeny-Aware Gap Placement Prevents Errors in Sequence Alignment and Evolutionary Analysis. Science. 320, 1632–1635 (2008).
>
> [4] - Löytynoja A, Goldman N. An Algorithm for Progressive Multiple Alignment of Sequences with Insertions. Proceedings of the National Academy of Sciences of the United States of America. (2005).
>
> [5] - Benner, Steven A., Mark A. Cohen, and Gaston H. Gonnet. “Empirical and Structural Models for Insertions and Deletions in the Divergent Evolution of Proteins.” Journal of Molecular Biology 229, no. 4 (1993).
>
> [6] - Chang, Mike S. S., and Steven A. Benner. “Empirical Analysis of Protein Insertions and Deletions Determining Parameters for the Correct Placement of Gaps in Protein Sequence Alignments.” Journal of Molecular Biology (2004).
>
> [7] - J, Zhang, Xiao L, Yin Y, Sirois P, Gao H, and Li K. “A Law of Mutation: Power Decay of Small Insertions and Small Deletions Associated with Human Diseases.” Applied Biochemistry and Biotechnology (2010).
>
> [8] - Gu, X., and W. H. Li. “The Size Distribution of Insertions and Deletions in Human and Rodent Pseudogenes Suggests the Logarithmic Gap Penalty for Sequence Alignment.” Journal of Molecular Evolution (1995).
>
> [9] - Katoh K, Kuma K, Toh H, Miyata T. MAFFT version 5: improvement in accuracy of multiple sequence alignment. Nucleic Acids Res (2005).
>
> [10] - R. C. Edgar, MUSCLE: multiple sequence alignment with high accuracy and high throughput. Nucleic Acids Res. (2004).
>
> [11] - S. R. Eddy, Accelerated profile HMM searches. PLoS Comp. Biol., (2011).
>
> [12] - Steinegger M, Söding J. MMseqs2 enables sensitive protein sequence searching for the analysis of massive data sets. Nat Biotechnol. (2017).

---

> ### Author Response · Authors · 2022-11-18
> **Author Response to AUpf [1 / 2]**
>
> Dear Reviewer,
>
> First and foremost, we are grateful for receiving your detailed feedback.
>
>
> *Comment*: “It is an interesting idea, and the authors resolve the main obstacle of transformer sequence length limitation with segments.”
>
>
> We thank the reviewer for this comment, we find applying learning mechanism to the sequence alignment interesting as well. The reviewer points to a major technical aspect of the proposed methodology, dealing with long biological sequences. However, in our view, the main contribution of the paper is conceptual - applying a learning mechanism for the sequence alignment problem, which allows the aligner to fit the data without the need to use default models for all input sequences, which is clearly unrealistic.
>
>
> *Comment*: “The alignment comparison between BetaAlign and traditional ones shows that it is comparable, but it remains a question that why we need deep learning here to achieve similar results but consume more time”
>
>
> We hypothesize the paper is the first step in developing the next generation of sequence aligners. Despite the computational limits, we think the main breakthroughs are the possibilities to learn to align biological sequences from data. As stated in our paper, the main advantage is allowing the personalizing of the transformer for each alignment. BetaAlign is designed to align sequences where the data has characteristics that are not necessarily captured well by current aligners. BetaAlign fits data that may come from specific evolutionary processes, governing indel rates, indel distribution, conserved areas, substitutions matrix, etc’. It will be best suited for labs where the accuracy of each column is critical. Future steps will include complex evolutionary events that are currently not issued in other datasets or aligners (even if there is a vast knowledge of their occurrences). Please have a look at our new section: “Biological Plausible Sequences”.
>
>
> *Comment*: “Due to the nature of the transformer, there is no guarantee that all input(e.g. ATCG) will be mapped to the output”
>
>
> Adding a constraint to force the transformer to generate only valid alignments is an interesting approach that we have considered but chose not to further investigate. Our approach to training ensembles of transformers has proven itself effective (see Figure 12). Training multiple transformers for each dataset has several more advantages such as introducing alignment uncertainty.
>
>
> *Comment*: “Datasets based on structural alignment should be used as the benchmark dataset, like BAliBASE”
>
>
> Several papers over the last decade have suggested some problems in the correctness of the manually computed alignment datasets that are considered the “gold standard” ([1], [2]). Usually, those datasets are aligned manually and hence do not follow the latest papers regarding the indel distributions. To overcome this hurdle, we choose to generate the misspecification dataset (see section “Model Misspecification”). Evaluating different aligners' performance on such datasets (BaliBase RV912 and RV911) has low accuracy which raises the question if labels are correct or the aligners. Even the latest aligners haven’t demonstrated the results on those databases [3], [4].

---

### Official Review · Reviewer_JNkr · 2022-10-29

**Confidence:** 3
**Correctness:** 3
**Technical Novelty And Significance:** 2
**Empirical Novelty And Significance:** 3
**Recommendation:** 6

**Clarity, Quality, Novelty And Reproducibility:**

Please see above. Overall, I think the paper is short on technical novelty but is applied to a novel domain. The results are interesting and supportive of the approach but the paper needs more analysis.

**Strength And Weaknesses:**

I am not well-versed in computational biology, so I am unable to judge the quality of empirical analysis because I am not sure how effectively the chosen datasets reflect progress in this area, and I am also unaware of the technical details of the baselines and models that are used in comparison. Hence, I will mostly reserve my detailed comments on the proposed technique and the results that are described in the paper.

Strengths:

-- The approach is fairly straightforward to implement, and seems reasonable to try.

-- The results show that the proposed approach is in general better than the baselines when considering the performance across various numbers of sequences to be aligned. Although MUSCLE, and PRANK consistently perform similarly to the proposed approach.

-- The "model misspecification" experiment is interesting because it aims to characterize robustness to distributional variaitons in training/test data. The proposed approach does seem robust to the tested misspecification.

Weaknesses:

-- The approach involves several heuristics and is a fairly common instantiation of generic transformer-based sequence to sequence paradigm. Hence, it is low on technical novelty. However, the application domain of computational biology is unusual and I haven't seen the use of this paradigm in the proposed domain.

-- The main contribution of the paper is to develop schemes for flattening multiple sequences into one input sequence and use the aligned sequence as the output sequence. Although, two such schemes are compared, the choice of the scheme seems arbitrary and inadequate analysis/ theoretical justification is provided for the choice.

-- I might be mistaken but it seems like the test data is also synthetically generated. While this would still be fine for comparing approaches, performance on naturally occurring data would strengthen the paper. Also, why not generate test data under all the proposed schemes in Table 2 and measure performance on different kinds of datasets?

-- The paper would benefit from a detailed description of the baselines in the main text so that it is clear how the proposed approach differs from existing approaches.

-- Similar to the point above, more analysis to show how the proposed approach is doing better compared to the baselines would help. Concretely, characterizing the errors various approaches differ in would significantly strengthen the paper. Do the baselines have hyperparameters that could be tuned? Were they tuned for different number of sequences etc.

-- MUSCLE and PRANK seem fairly close to the proposed approach. Interestingly, while the results show them to be competitive/better for neucleotides and worse for protein, the misspecification experiment flips this order. This is interesting and should be studied in detail.

-- Minor point: Instead of using "words", please use "token" while talking about sequences.

**Summary Of The Paper:**

This work uses sequence-to-seqeunce transformers to align multiple sequences which is an important problem in computational biology.  The approach mainly proposes a scheme to convert multiple seqeunces into one long input sequence and the desired output aligned sequence as the output sequence in the seq2seq setup. The training data for this setup is generated via simulation. To account for long sequences, a heuristic scheme for splitting inputs/outputs into multiple shorter seqeunces is described. An ensemble of transformers is trained where each transformer is trained on the data from a different simulation and this ensemble is used for prediction. Finally, the proposed approach is compared against other commonly used multiple sequence aligners on datasets of neucleotide and protein sequences.

**Summary Of The Review:**

The proposed approach is reasonable and the empirical comparison shows its superiority over other baselines. However, more analysis needs to be done and the baselines and experiments should be more thoroughly explained.

---

> ### Author Response · Authors · 2022-11-18
> **Author Response to JNkr [2 / 2]**
>
> *Comment*: “Do the baselines have hyperparameters that could be tuned? Were they tuned for different number of sequences etc”
>
> The baselines do have hyperparameters but mostly the default is used. There is usually a gap between those parameters to the assumptions you have on the data, which makes it difficult to tune them. In addition, the baselines are not flexible and adding an assumption requires complex modifications (e.g., the differentiation of the deletion from the insertion at PRANK [3]). We hypothesize those are the main drawbacks regarding the future of progressive multiple alignment, as they don’t scale well regarding new evolutionary information.
>
>
> *Comment*: “Also, why not generate test data under all the proposed schemes in Table 2 and measure performance on different kinds of datasets”
>
> We choose to present only the best scheme out of those we tested. Evaluating all of the schemes for each of the datasets is computationally intensive.
>
>
> If you have any more comments, please let us know.
>
>
> [1] - David A. Morrison, Why Would Phylogeneticists Ignore Computerized Sequence Alignment?, Systematic Biology, (2009).
>
> [2] -  Iantorno S. Gori K. Goldman N. Gil M. Dessimoz C. Who watches the watchmen? An appraisal of benchmarks for multiple sequence alignment Methods Mol. Biol. (2014).
>
> [3] - A. Löytynoja, N. Goldman, Phylogeny-Aware Gap Placement Prevents Errors in Sequence Alignment and Evolutionary Analysis. Science. 320, 1632–1635 (2008).

---

> ### Author Response · Authors · 2022-11-18
> **Author Response to JNkr [1 / 2]**
>
> Dear Reviewer,
>
> First and foremost, we appreciate your detailed feedback.
>
> *Comment*: “...the application domain of computational biology is unusual and I haven't seen the use of this paradigm in the proposed domain”
>
> We thank the reviewer for this comment. Indeed, the vast majority of the data in biology are sequences (amino acids and nucleotides), and thus we expect that in the near future additional research groups will utilize NLP tools for computational biology. Currently, the application of NLP to such task is very novel.
>
> *Comment*: “The main contribution of the paper is to develop schemes for flattening multiple sequences into one input sequence and use the aligned sequence as the output sequence”
>
> The reviewer points to a major technical aspect of the proposed methodology. However, in our view, the main contribution of the paper is conceptual - applying a learning mechanism for the sequence alignment problem, which allows the aligner to fit the data without the need to use default models for all input sequences, which is clearly unrealistic. Moreover, as stated in the paper, there is a vast knowledge of different evolutionary processes which are not considered in today’s aligners, e.g., the probability of opening a gap depends on the sequence context. All current approaches do not take this sequence context into account. Our approach, in contrast implicitly does. This is an elegant and apparently easier tool to allow an aligner to consider such knowledge from training data. In the revised version we better emphasize these contributions in the “Biological Plausible Sequences” section.
>
> *Comment*: “...While this would still be fine for comparing approaches, performance on naturally occurring data would strengthen the paper”
>
> The main problem with evaluating performance on naturally data in the sequence alignment task is that the correct label is unknown [1], [2]. Thus, it is common in our field to evaluate the performance on simulated data [3].
>
> *Comment*: “The paper would benefit from a detailed description of the baselines in the main text so that it is clear how the proposed approach differs from existing approaches”
>
> We added a “related work” section in the revised paper. Our approach is drastically different from previous concepts of aligners. We don’t consider any scoring scheme but learn it straight from the data. This allows us to consider different types of indel distributions and different rates for insertion and deletion. In addition, we are inferring directly the multiple sequence alignment based on all of the sequences instead of using pairs. This allows the transformer to have the entire data at the start of the alignment process.
>
> *Comment*: “Similar to the point above, more analysis to show how the proposed approach is doing better compared to the baselines would help. ”
>
> We have generated a dataset with different evolutionary pressures on different places over the amino-acid sequences. One might think of this dataset as a protein with different selection pressures. There are areas with low, medium and high indel rates. Those areas refer to conserved and nonconserved areas. BetaAlign has improved the accuracy compared to the different aligners. Please see the new section “Biological Plausible Sequences”.
>
> *Comment*: “Concretely, characterizing the errors various approaches differ in would significantly strengthen the paper.”
>
> We appreciate this comment as it helped us demonstrate the implicit assumptions of the baseline aligners. We conducted another analysis characterizing the errors of the different aligners. Please see the new section “Characterizing the Error of Different Aligners”. The results clearly show the superiority of BetaAlign with the extracted features (indel focused). The baseline aligners are especially biased when long indels occur.

---

### Decision · Program_Chairs · 2023-01-20

**Decision:**

Accept: poster

**Justification For Why Not Higher Score:**

There are too many small issues like a lack of a more comprehensive out-of-distribution analysis, lacking error analysis, and a lack of highlighting some of the weaknesses for the paper to be highlighted at the conference.

**Justification For Why Not Lower Score:**

see meta review.

**Metareview: Summary, Strengths And Weaknesses:**

The paper proposes a novel solution to the sequence-to-sequence alignment problem based on transformer. The sequence-to-sequence alignment problem is an important problem in computational biology and is often solved with dynamic programming approaches. The paper proses a creative solution to this problem by formulating the sequence-to-sequence alignment problem as a problem of translating from one to another language, which in turn is addressed by transformers.

I find this is a very interesting idea, and enjoyed reading the paper very much. The idea to formulate the sequence-to-sequence alignment problem as a translation problem is very creative. This is also the first approach that I have seen that uses deep networks for sequence alignment, and for this it yields very competitive results. Therefore I think this paper is a strong addition to the literature on `learning algorithms'.

However, all reviewer raised the point that the network is trained on simulated data, and application on real data would be preferable. The authors raised convincing arguments for their choice of using simulated data, and I don't see training and evaluating  as a major weakness. However, I recommend to explain the reasons for using simulated data well in the final paper, since this will be a question many readers from the machine learning community will have.

A more significant issue raised by R2 is that the training and test distributions are equal or are very similar. Thus, we don't get a good idea on how the method works on data that differs significantly from the training distribution, and this would be very important to know. I therefore recommend that the authors add such simulation in their final paper, and discuss this in a limitations section.

Another issue raised by R1 and R2 is that an aligned output sequence could be different from the input sequence. This failure case can easily be detected, but this is an important point, and I suggest the paper makes this very clear in the final paper that this is a potential limitation.




**Note From Pc:**

if the above contains the word "oral" or "spotlight" please see: "oral" presentation means -> notable-top-5% and "spotlight" means -> notable-top-25%. As stated in our emails, we are disassociating presentation type from AC recommendations

**Summary Of Ac-Reviewer Meeting:**

Below I summarize how I weighted the different points raised in the virtual in-person discussion and in the reviews.

R1, an NLP expert, notes that the method is shown to perform better or equal to baseline methods such as MUSCLE.
As weaknesses, the reviewer notes that the approach uses several heuristics and is a common instantiation of a transformer, and does therefore not have much technical novelty. The authors respond that the novelty lies in the formulation of the problem as a sequence-to-sequence translation problem. In my opinion, this formulation is not obvious, and thus there is technical novelty in formulating the problem so that it can be addressed by transformers.

Another weakness mentioned by R1 is that the test data is synthetically generated. The authors respond that in this field, data is typically simulated since no ground truth is available.

The reviewer proposesd to characterize the errors made by the different approaches, and the authors provided this information.

R2 raised the following concerns:

- There is no guarantee that input tokens are not corrupted, i.e., a output sequence could be different from the input sequence. This failure case can easily be detected, but this is an important point, and I suggest the paper makes this potential limitation clear in the final version of the paper.

- Why use deep learning when it's not faster or better? The authors argue that this is a first step, and is conceptually interesting. I agree.

- R2 also notes that simulated data is used, and using simulated data as train and test data is not fair, and suggests a concrete dataset to evaluate on. The authors also evaluate on somewhat out-of-distribution data, and they note that they constructed the data carefully, following biologically plausible processes, and that it is common in the community to compare on simulated data.

R3 notes that the paper likely improves the SOTA for some applications. R3 also notes that the computational cost can be prohibitive. The training is indeed costly, but the method can be relatively fast at inference and is thus potentially useful in my opinion.